# PROMPTING THE UNSEEN: DETECTING HIDDEN BACKDOORS IN BLACK-BOX MODELS

## ABSTRACT

Visual prompting (VP) is a new technique that adapts well-trained frozen models for source domain tasks to target domain tasks. This study examines VP's benefits for black-box model-level backdoor detection. The visual prompt in VP maps class subspaces between source and target domains. We identify a misalignment, termed class subspace inconsistency, between clean and poisoned datasets. Based on this, we introduce BPROM, a black-box model-level detection method to identify backdoors in suspicious models, if any. BPROM leverages the low classification accuracy of prompted models when backdoors are present. Extensive experiments confirm BPROM's effectiveness.

## 1 INTRODUCTION

Deep neural networks (DNNs) are commonly used in complex applications but require extensive computational power, leading to significant costs. Users often access these models through online platforms like BigML model market[1] and ONNX zoo[2], or via Machine Learning as a Service (MLaaS) platforms. However, DNNs can include backdoors (Gu et al., 2017; Liu et al., 2018b; Tang et al., 2021; Qi et al., 2023b; Nguyen & Tran, 2021; Chen et al., 2017), which manipulate model responses to inputs with specific triggers (like certain pixel patterns) while functioning correctly on other inputs. In backdoor attacks, attackers embed these triggers in the training data, leading the model to associate the trigger with a particular outcome and misclassify inputs containing it.

**Why Black-Box Model-Level Detection.** Black-box backdoor detection, which uses only black-box queries to the suspicious model (i.e., the model to be inspected), is gaining attention. This detection method is divided into input-level (Li et al., 2021c; Qiu et al., 2021; Gao et al., 2022; Liu et al., 2023; Qi et al., 2023c; Zeng et al., 2023; Guo et al., 2023; Hou et al., 2024; Xu et al., 2024; Mo et al., 2024) and model-level (Huang et al., 2020; Dong et al., 2021; Guo et al., 2022; Xu et al., 2019; Wang et al., 2024) techniques. Input-level detection identifies trigger samples in an infected model, while model-level detection determines if a model contains backdoors. Input-level detection relies on the model having backdoors; otherwise, its accuracy drops significantly. For example, as shown in Table 1, TeCo (Liu et al., 2023) and SCALE-UP (Guo et al., 2023), state-of-the-art input-level detectors, show AUROCs of 0.8113 and 0.7877, respectively, on a BadNets-infected model (Gu et al., 2017), but only 0.4509 and 0.5103 on a clean model. If a model is clean, many legitimate samples may be misclassified as triggers, reducing the model's practical utility. Thus, model-level detection should be performed first. If backdoors are found but the model must still be used, input-level detection should then be applied to each input.

**Design Challenge.** Despite its importance, black-box model-level detection faces two main challenges. First, unlike input-level detection, which benefits from the presence of an infected model, model-level detection has limited ground truth, relying on only a few clean samples. Second, it needs a stable feature to differentiate between clean and infected models across various backdoor types, which is difficult to find. For instance, B3D (Dong et al., 2021) targets trigger localization but is mainly effective for patch-based triggers. Similarly, AEVA (Guo et al., 2022) may struggle with larger triggers due to its dependence on adversarial peak analysis.

**Our Design.** Visual prompting (VP) (Bahng et al., 2022; Jia et al., 2022) allows a frozen, pre-trained model from a source domain to correctly predict samples from a target domain by applying a visual

---

[1] https://bigml.com/
[2] https://github.com/shaoxiaohu/model-zoo

Table 1: A significant drop of F1-score and AUROC in black-box input-level detection methods, TeCo (Liu et al., 2023) and SCALE-UP (Guo et al., 2023).

| TeCo (Liu et al., 2023) | BadNet (Gu et al., 2017) | | Blended (Chen et al., 2017) | | WaNet (Nguyen & Tran, 2021) | |
|---|---|---|---|---|---|---|
| | Backdoored | Clean | Backdoored | Clean | Backdoored | Clean |
| F1 | 0.8014 | 0.5263 | 0.7621 | 0.5033 | 0.9295 | 0.5137 |
| AUROC | 0.8113 | 0.4509 | 0.7259 | 0.3954 | 0.9345 | 0.4406 |
| ScaleUp (Guo et al., 2023) | Backdoored | Clean | Backdoored | Clean | Backdoored | Clean |
| F1 | 0.7964 | 0.5236 | 0.7991 | 0.5046 | 0.7199 | 0.4768 |
| AUROC | 0.7877 | 0.5103 | 0.7694 | 0.4643 | 0.7772 | 0.4246 |

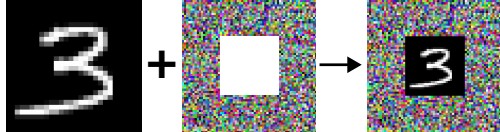 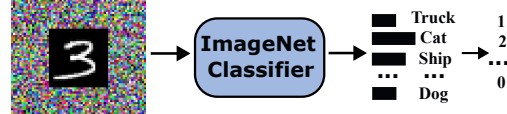

(a) The "3" is from MNIST, while the middle part shows the visual prompt. The prompted sample is ready for ImageNet classifier.

(b) The prompted sample can be fed into the ImageNet classifier, whose output has a mapping between the labels from MNIST and ImageNet.

Figure 1: How a frozen ImageNet classifier is adapted for the MNIST classification when VP is used.

prompt. This technique can work across very different domains; for example, an ImageNet classifier (source) can detect melanoma (target) via VP (Tsai et al., 2020). Figure 1 illustrates VP, where the visual prompt (trainable noise in Figure 1a) maps between class subspaces of the source and target domains, enabling the frozen classifier to handle the target task efficiently.

In an infected model, the target class subspace in the feature space is adjacent to all other class subspaces (Wang et al., 2019). We identify a *class subspace inconsistency* where misalignment between class subspaces in the poisoned (source) and clean (target) datasets leads to low classification accuracy of the prompted model. This phenomenon is illustrated in Figure 2 and experimentally validated in both Figure 3 and Section C. Based on this, we propose BPROM for black-box model-level backdoor detection. BPROM applies VP to a suspicious model using an unrelated clean dataset; poor accuracy in the prompted model indicates the presence of backdoors.

**Contribution.** Our contributions can be summarized as follows. 1) We identify a *class subspace inconsistency* in VP on backdoor-infected models. This misalignment between class subspaces of the poisoned dataset and an external clean dataset signals backdoor infection. 2) Utilizing this inconsistency, we develop BPROM, a black-box model-level backdoor detection method.

## 2 RELATED WORKS

We do not aim to provide a comprehensive review of backdoor attacks and defenses; for a detailed survey, see (Li et al., 2022).

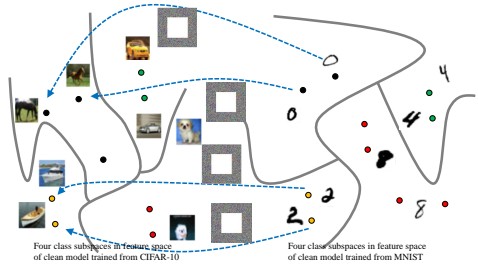 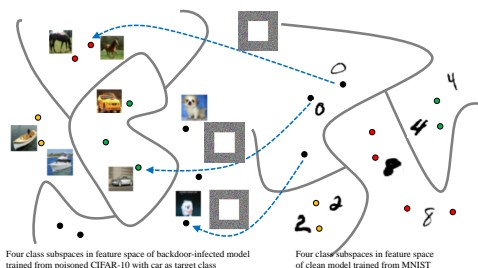

(a) Class subspace inconsistency does not occur: visual prompt as a mapping between two clean datasets.

(b) Subspace inconsistency occurs: visual prompt as a mapping between clean and poisoned datasets.

Figure 2: A conceptual illustration of (a) VP on clean model and (b) VP on backdoor-infected model.

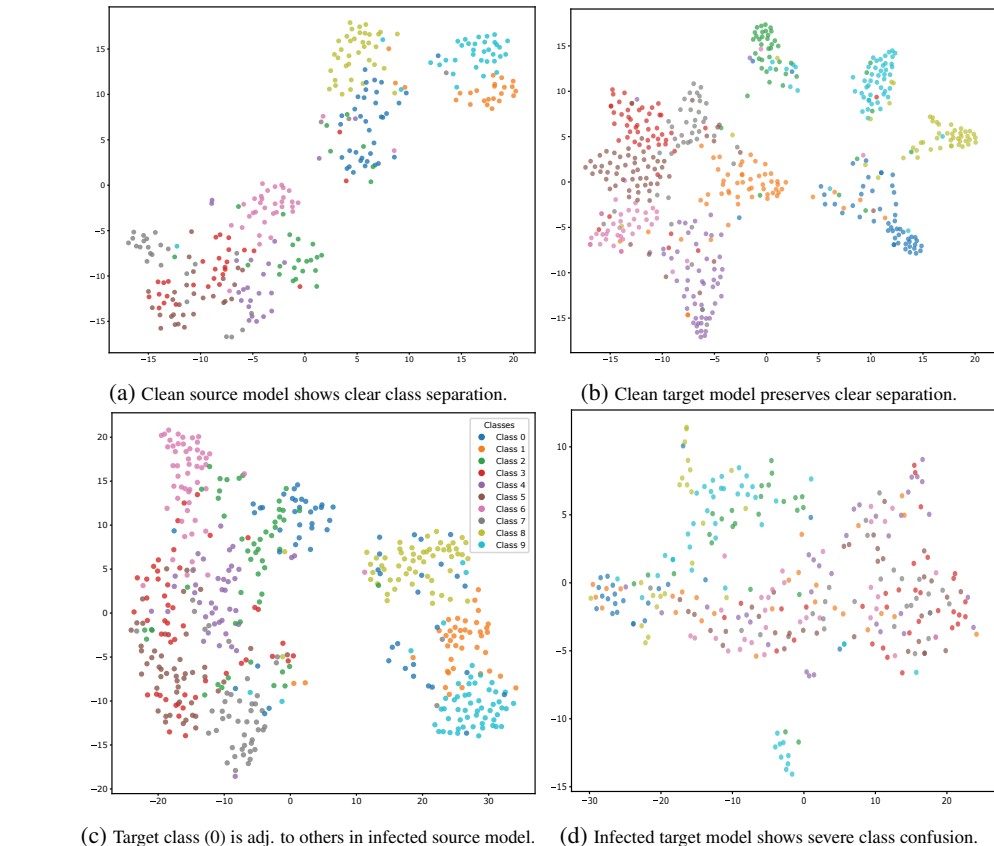

(a) Clean source model shows clear class separation.

(b) Clean target model preserves clear separation.

(c) Target class (0) is adj. to others in infected source model.

(d) Infected target model shows severe class confusion.

Figure 3: Class subspaces inconsistency (CIFAR-10 for source model and STL-10 for target model).

**Backdoor Attack Methods.** Badnets (Gu et al., 2017) introduced the first backdoor attack on DNNs, with many following works adopting its approach to poison training datasets. Backdoor attacks are categorized by trigger appearance into universal (Chen et al., 2017; Gu et al., 2017; Zeng et al., 2021), where all triggers are identical, and sample-specific (Li et al., 2020; Nguyen & Tran, 2020; Salem et al., 2022), where triggers vary per sample. Subsequent developments include invisible backdoors (Doan et al., 2021; Li et al., 2020; Nguyen & Tran, 2021), which are harder to detect by human inspection, and clean-label backdoors (Zhao et al., 2020; Ning et al., 2021; Shafahi et al., 2018; Turner et al., 2018), which stealthily poison target class samples without label changes. Additionally, anti-defense attacks (Qi et al., 2023b) circumvent detection by preventing latent separation.

**Backdoor Detection Methods.** Backdoor detection is categorized into white-box and black-box. White-box detection (Liu et al., 2018a; Wu & Wang, 2021; Li et al., 2021a; Xia et al., 2022; Du et al., 2020; Li et al., 2021b; Huang et al., 2022; Wang et al., 2019; Hu et al., 2022; Tao et al., 2022; Wei et al., 2024; Li et al., 2023a; Wang et al., 2024) requires access to a poisoned training set or model parameters. Some methods identify backdoors, while others remove them. However, it is unsuitable for MLaaS applications and safety-critical deployments (e.g., autonomous vehicles).

Black-box detection only requires access to the suspicious model, making it more applicable. It is divided into input-level and model-level. Input-level detection (Li et al., 2021c; Qiu et al., 2021; Gao et al., 2022; Liu et al., 2023; Qi et al., 2023c; Zeng et al., 2023; Guo et al., 2023; Xian et al., 2024; Ma et al., 2022; Pan et al., 2023; Jin et al., 2022; Chen et al., 2024; Zhu et al., 2024; Hou et al., 2024; Xu et al., 2024) distinguishes trigger samples from benign ones. Since infected models act benign except for trigger samples, they can be used safely if detection works per input. However, this can result in high false positives, rejecting many benign samples if the model is clean, as shown in Table 1.

This paper focuses on model-level detection (Huang et al., 2020; Dong et al., 2021; Guo et al., 2022; Xu et al., 2019; Shi et al., 2024; Xiang et al., 2024; Sun et al., 2023; Rezaei et al., 2023; Wang et al.,

2024), which identifies backdoors in suspicious models and serves as front-line detection before input-level methods.

# 3 BACKGROUND KNOWLEDGE

Both visual prompting (VP) (Chen et al., 2023; Bahng et al., 2022; Jia et al., 2022) and model reprogramming (MR) (Tsai et al., 2020; Chen, 2024; Elsayed et al., 2019; Neekhara et al., 2022) enable a frozen pre-trained model for one task to perform a different target domain classification task by deriving a visual prompt for inputs from the target domain. Initially, MR was considered an *attack* that misused cloud services (i.e., MLaaS) to perform undocumented tasks (Elsayed et al., 2019). VP was recently introduced in (Bahng et al., 2022). Although VP and MR share the same concept, VP focuses exclusively on images. VP has been extended to image inpainting (Bar et al., 2022), antibody sequence infilling (Melnyk et al., 2023), and differentially private classifiers (Li et al., 2023b). In this paper, VP and MR are used interchangeably, with the visual prompt in VP corresponding to the trainable noise in MR. More formally, VP/MR proceeds with four steps (Chen, 2024).

1. Initialization: Let $f_S(\cdot)$ and $D_T = \{(x_T, y_T)\}$ be the source model (the model trained from the source domain dataset) and the target domain dataset, respectively. Randomly initialize $\theta$ and $w$ (defined below).

2. Visual prompt padding: Obtain the prompted input sample $\tilde{x}_T = V(x_T|\theta)$, where $\theta$ is the visual prompt. A common method for $V(\cdot)$ is to resize $x_T$ and add the visual prompt (trainable noise) around it. Although $\tilde{x}_T$ visually differs from the source domain, it can still be used as input for the source domain classifier. Figure 1a illustrates this with $x_T$ as "3" from MNIST, $\theta$ in the middle, and $V(\cdot)$ resizing $x_T$ and padding it with $\theta$.

3. Output mapping: Obtain the target task prediction via $\hat{y}_T = O(f_S(\tilde{x}_T)|w)$, where $w$ represents the trainable parameters for output label mapping. This step is optional for VP/MR. In our experiment, we omitted this step.

4. Prompted model training: Optimize $\theta$ and $w$ by minimizing a task-specific loss $\mathcal{L}(\hat{y}_T, y_T)$ on $D_T$.

After executing the four-step procedure, we obtain the prompted model $f_T = O \circ f_S \circ V$ from $f_S(\cdot)$ with optimized $\theta^*$ and optionally $w^*$. This results in $\hat{y}_T = O(f_S(V(x_T|\theta^*))|w^*)$.

# 4 SYSTEM MODEL

**Threat Model.** We consider two roles: attacker and defender. The attacker's goal aligns with previous work (Gu et al., 2017; Chen et al., 2017; Tang et al., 2021; Qi et al., 2023b; Liu et al., 2018b). Specifically, the attacker poisons the training dataset by injecting trigger samples. The DNN model (e.g., an image classifier) trained on this poisoned dataset behaves normally with clean inputs but always predicts an attacker-specified target class for inputs with a trigger. Essentially, an all-to-one backdoor is implanted, mapping all trigger inputs to a specific target class.

**Defender's Goal and Capability.** The defender's goal is to detect if a suspicious model is backdoored, primarily measured by AUROC (see Section 6). The defender has limited abilities: no access to the poisoned dataset, model structure, or parameters. In MLaaS applications, detection involves only black-box queries on the model to obtain confidence vectors. The defender also has a small reserved clean dataset $D_S$ (1%, 5%, 10% of the test dataset in our experiment) to aid detection.

# 5 PROPOSED METHOD

We present our detection method, BPROM. The notation table can be found in Table 27 in Appendix E.

## 5.1 OVERVIEW

Different clean datasets have distinct class subspace "shapes" in feature space. However, as noted in Wang et al. (2019), poisoned datasets exhibit target class subspaces that share boundaries with

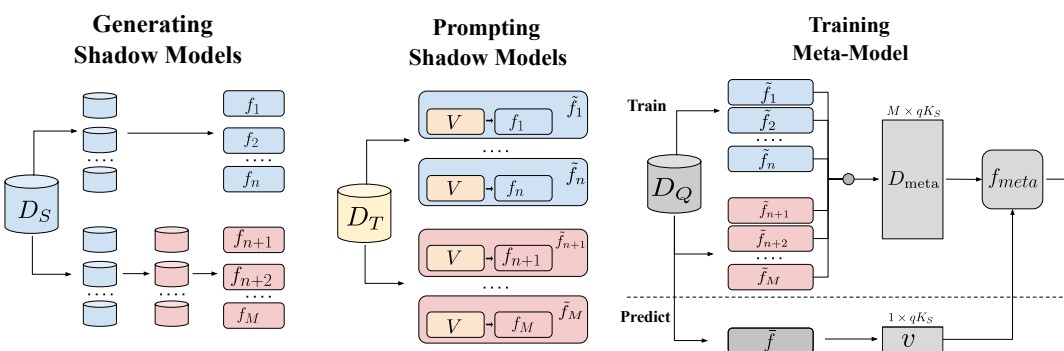

Figure 4: The workflow of BPROM. The blue and red components are related to $D_S$ and $D_P$, respectively. The yellow parts are related to VP and $D_T$. The gray components have connection to $D_Q$ and are used for train $f_{\text{meta}}$

.

all others. This creates misalignment when adapting a poisoned model to a clean dataset, termed *class subspace inconsistency*, resulting in reduced prompted model accuracy. This is conceptually illustrated in Figure 2 and experimentally validated in Section C and Figure 3. As an evidence, Table 2 also shows that an increasing number of target classes worsens the inconsistency (i.e., lower accuracy). BPROM leverages this for backdoor detection. The core idea is that adapting an infected source model to a clean target task via visual prompting is significantly harder due to the class subspace mismatch. Theorem 1 in Yang et al. (2021) states that target risk is bounded by source risk and representation alignment loss. For BPROM, this alignment loss is amplified by the inconsistency in infected models, leading to poor target task performance. Thus, low prompted accuracy signals potential backdoors. To achieve effective detection, the BPROM training has three steps: shadow model generation, prompting, and meta-model training. First, diverse poisoned and clean shadow models are trained. Second, visual prompts are learned for each shadow model using an external clean dataset. Finally, a meta-classifier is trained on confidence vectors from prompted shadow models to detect backdoors. The workflow and pseudocode are shown in Figure 4 and Algorithm 1.

## 5.2 BPROM

**Generating Shadow Models.** The goal of this step is to construct shadow models, categorized into clean and backdoor shadow models. Clean shadow models are trained on a clean dataset, while backdoor shadow models are trained on a poisoned dataset.

Let $D_S$ be the reserved clean dataset. To check if a suspicious model was trained on CIFAR-10, $D_S$ includes a limited number of CIFAR-10 samples (e.g., 1%, 5%, 10% in our experiment). The defender trains $n$ clean shadow models, $f_i$'s, with different parameter initializations. Given a poisoning rate $p$ and a chosen backdoor attack, the defender creates $M - n$ poisoned datasets by injecting trigger samples according to the chosen attacks, where $M$ is the total number of shadow models. Specifically, each poisoned dataset $D_P$ is constructed as follows:

Table 2: Class subspace inconsistency worsens (i.e., the prompted model's testing accuracy decreases) as the number of target classes increases.

| # target classes | 1 | 2 | 3 |
|---|---|---|---|
| CIFAR10 | 0.3286 | 0.2427 | 0.2338 |
| GTSRB | 0.2711 | 0.1988 | 0.1986 |

**Step 1:** A proportion $p$ of samples $(x, y)$ from the clean dataset $D_S$ are extracted to form $D_E$.

**Step 2:** The extracted samples are transformed by adding a trigger pattern $(m, t, \alpha, y_t)$ to obtain poisoned counterparts $\{(x', y')|x' = (1 - m) \cdot x + m \cdot ((1 - \alpha)t + \alpha x), y' = y_t\}$, where $y_t, m, t, \alpha$, $\cdot$ denote the target class, trigger mask, trigger, intensity, and element-wise product, respectively (Guo et al., 2022; 2023).

**Step 3:** Construct $D_P = (D_S \setminus D_E) \cup \{(x', y')\}$. By sampling different combinations of backdoor patterns $(m, t, \alpha, y_t)$, various $D_P$ can be generated. Backdoor shadow models are trained on $D_P$'s.

**Prompting Shadow Models.** This step applies VP to both types of shadow models (clean and poisoned) to generate prompted shadow models. Let $D_T = D_T^{\text{train}} \cup D_T^{\text{test}}$ be an external clean dataset, with $D_T^{\text{train}}$ as the training set and $D_T^{\text{test}}$ as the test set. $D_T$ can have a different distribution than $D_S$. For shadow models, prompts ($\theta_i$) are learned via standard backpropagation on $D_T^{\text{train}}$. This process is also applied to the suspicious model $f_{sus}$, but using a gradient-free optimization method (e.g., CMA-ES) since we only have black-box access. This results in the prompted shadow models $\tilde{f}_i(\cdot) = f_i(V(\cdot|\theta_i^*))$, and prompted suspicious model $\bar{f}(\cdot) = f_{sus}(V(\cdot|\theta_{sus}^*))$. Detailed steps for VP can be found in Section 3 (e.g., (Bahng et al., 2022)).

**Meta Model Training.** The goal of this step is to train a binary classifier $f_{\text{meta}}$ for backdoor detection. For each shadow model $\tilde{f}_i$, the defender randomly selects $q$ samples from $D_T^{\text{test}}$ to form $D_Q = \{x_Q^1, \ldots, x_Q^q\}$. Each sample from $D_Q$ is fed to $\tilde{f}_i$. The defender creates a dataset $D_{\text{meta}} = D_{\text{meta}} = \{(\tilde{f}_i(x_Q^1)||\cdots||\tilde{f}_i(x_Q^q), \text{clean})\}_{i=1}^n \cup \{(\tilde{f}_i(x_Q^1)||\cdots||\tilde{f}_i(x_Q^q), \text{backdoor})\}_{i=n+1}^M$. Here, $\tilde{f}_i(x_Q^j)$ is the confidence vector, and its length, $K_S$, is the number of classes in $D_S$. The defender then trains a binary classifier $f_{\text{meta}}$ using $D_{\text{meta}}$.

**Backdoor Detection on Suspicious Model.** To inspect a suspicious model $f_{sus}$, we first obtain $q$ confidence vectors from the prompted suspicious model $\bar{f}$. These vectors are concatenated and fed to $f_{\text{meta}}$. Specifically, $v = (\bar{f}(x_Q^1)||\cdots||\bar{f}(x_Q^q))$ is computed and input to $f_{\text{meta}}$, which outputs either clean or backdoor.

## 5.3 Discussion

BPROM is similar to MNTD (Xu et al., 2019), but they have important differences.

**More Efficient Data Generation:** In BPROM, the defender uses a single backdoor attack to generate $D_P$, whereas MNTD uses multiple backdoor attacks. Even if multiple methods are used in BPROM, detection accuracy improves only marginally. MNTD needs to "see" various backdoor types to better detect unknown backdoors. However, BPROM focuses on class subspace inconsistency, where $D_P$ learns different feature space partitions, with the target class adjacent to all other classes.

**Much Fewer Shadow Models Required:** BPROM needs only a few shadow models (e.g., 20 in our experiments), while MNTD requires hundreds due to the variety of backdoor attacks (e.g., 256 in MNTD). This reduces training costs and allows BPROM to achieve high performance (1.0 AUROC on CIFAR-10 for both BadNets and Blend, compared to MNTD's 0.92 and 0.955) even with a single backdoor type. Training MNTD is also much more complex than training BPROM.

**Novel Design Principle:** Most importantly, their design principles differ fundamentally. MNTD relies on meta-learning and needs to "see" various backdoor properties. BPROM relies on class subspace inconsistency, achieving decent detection accuracy (e.g., 0.8137 F1-score on CIFAR-10 with BadNets and STL-10, and 0.7499 with GTSRB and STL-10) even with a single shadow model and no reserved clean samples. The auxiliary design with a similar MNTD structure further improves performance.

---

**Algorithm 1:** BPROM.

**Input:** $D_S$ and $D_T = D_T^{\text{train}} \cup D_T^{\text{test}}$;
**Output:** $f_{\text{meta}}$

1  /* Generating Shadow Models    */
2  **for** $i = 1$ **to** $M$ **do**
3     Copy $D_S$ into $D_S^i$
4     **if** $i \leq n$ **then**
5        train $f_i$ from $D_S^i$
6     **else**
7        augment $D_S^i$ with triggers
8        train $f_i$ from the augmented $D_S^i$

9  /* Prompting Shadow Models    */
10 **for** $i = 1$ **to** $M$ **do**
11    learn visual prompt $\theta_i$ on $D_T^{\text{train}}$
12    construct $\tilde{f}_i = f_i \circ V(\cdot|\theta)$

13 /* Training Meta Model        */
14 Construct $D_Q = \{x_Q^1, x_Q^2, ..., x_Q^q\}$ by randomly sampling $q$ samples from $D_T^{\text{test}}$
15 Initialize $D_{\text{meta}}$ as an empty set
16 **for** $i = 1$ **to** $M$ **do**
17    **if** $i \leq n$ **then**
18       $v_i \leftarrow (\tilde{f}_i(x_Q^1)||\cdots||\tilde{f}_i(x_Q^q))$
19       $l_i \leftarrow$ 'clean'
20       $D_{\text{meta}} = D_{\text{meta}} \cup \{(v_i, l_i)\}$
21    **else**
22       $v_i \leftarrow (\tilde{f}_i(x_Q^1)||\cdots||\tilde{f}_i(x_Q^q))$
23       $l_i \leftarrow$ 'backdoor'
24       $D_{\text{meta}} = D_{\text{meta}} \cup \{(v_i, l_i)\}$

25 Train the binary classifier $f_{\text{meta}}$ using $D_{\text{meta}}$
26 **return** $f_{meta}$

---

## 6    EXPERIMENTS

We overview the experimental setup, including datasets, model architectures, attack methods, and defense baselines, consistent with recent works (Qi et al., 2023c; Guo et al., 2023; Huang et al., 2020; Liu et al., 2023). We then present the experimental results and hyperparameter study.

### 6.1    EXPERIMENTAL SETUP

**Datasets and Model Architectures.**    We use five image datasets: CIFAR-10 (Krizhevsky, 2009), GTSRB (Stallkamp et al., 2011), and STL-10 (Coates et al., 2011), Tiny-ImageNet (Le & Yang, 2015), and ImageNet (Russakovsky et al., 2015). For a suspicious model $f_S(\cdot)$ trained on CIFAR-10, GTSRB, Tiny-ImageNet or ImageNet, we first train shadow models $f_i$'s using an $\alpha\%$ ($\alpha \in \{1, 5, 10\}$) subset of the corresponding test set as $D_S$. Then, we apply VP on $f_i$'s using STL-10 as $D_T$ to obtain the corresponding prompted models $\tilde{f}_i$'s. We experiment with ResNet18 and MobileNetV2 architectures, training models on each $D_S^i$ and $D_T^{\text{train}}$ using standard procedures. For the meta-classifier $f_{\text{meta}}$, we use a random forest with 10,000 trees to detect backdoors based on confidence vectors. We mainly use Area Under the ROC Curve (AUROC) and F1-score to measure the detection effectiveness of backdoor detection methods. Our experiments were performed on a workstation equipped with a 16-core Intel i9 CPU (64GB RAM) and an RTX4090 GPU.

**Attack Methods and Defense Baselines.**    We evaluate BPROM against 9 backdoor poisoning attacks from the Backdoor Toolbox[3], including classical dirty label, clean label, sample-specific trigger, and adaptive attacks. Default hyperparameters are used to ensure at least 98% attack success rate. We compare BPROM with 10 backdoor defenses either from Backdoor Toolbox or from their official code. Default hyperparameters are used for each defense.

### 6.2    EXPERIMENTAL RESULT

We know from class subspace inconsistency that a prompted model's accuracy degrades if the suspicious model is backdoored. We conducted experiments with backdoor attacks using varying trigger sizes ($4 \times 4$, $8 \times 8$, $16 \times 16$ pixels) and poisoning rates (5%, 10%, 20% of training data) to further examine the impact of class subspace inconsistency on prompted model accuracy. For each experiment, we generated a backdoor-infected model and prompted it for a new task on STL-10. These experiments also cover adaptive attacks, where BPROM maintains high performance, achieving an AUROC of 1 even at low poison rates (e.g., 0.2% for BadNets on CIFAR-10; see Section 6.4).

**Trigger Size Impact.** Table 3 shows the accuracy of prompted models on STL-10 with varying trigger sizes. We trained backdoored models on CIFAR-10 and GTSRB, then prompted them to classify STL-10. As trigger size increases, accuracy decreases. This is because larger triggers distort feature representations more, worsening class subspace inconsistency.

Table 3: Testing accuracy for different trigger sizes.

|  | CIFAR-10 | | GTSRB | |
|---|---|---|---|---|
|  | Blend | Adap-Blend | Blend | Adap-Blend |
| (4*4) | 0.3830 | 0.3336 | 0.1783 | 0.1245 |
| (8*8) | 0.3517 | 0.3250 | 0.1641 | 0.1183 |
| (16*16) | 0.3172 | 0.3127 | 0.1571 | 0.1080 |

**Poison Rate Impact.** Table 4 shows the accuracy of prompted models with varying poison rates. Similar to the trigger size experiments, we trained backdoored models on CIFAR-10 and GTSRB, then prompted them for STL-10. Higher poison rates lead to lower accuracy due to increased feature distortion, consistent with our class subspace inconsistency explanation. Both Table 3 and Table 4 show low accuracies, supporting this reasoning.

Table 4: Testing accuracy for various poison rates.

|  | CIFAR-10 | | GTSRB | |
|---|---|---|---|---|
|  | Blend | Adap-Blend | Blend | Adap-Blend |
| 5% | 0.5297 | 0.5233 | 0.2488 | 0.2368 |
| 10% | 0.4772 | 0.4830 | 0.2328 | 0.2036 |
| 20% | 0.3985 | 0.3358 | 0.2222 | 0.1705 |

**Performance on CIFAR-10 and GTSRB Baselines.** Table 25 compares defenses using ResNet18 as the shadow and suspicious model (infected ResNet18 has accuracy $> 0.92$ and attack success

---

[3]https://github.com/vtu81/backdoor-toolbox

Table 5: Area Under the ROC Curve (AUROC) of defenses on ResNet18 with different datasets. AVG stands for the average AUROC. Green (red) cells denote values greater (lower) than 0.8.

| | | Badnets (Gu et al., 2017) | Blend (Chen et al., 2017) | Trojan (Liu et al., 2018b) | BPP (Wang et al., 2022) | WaNet (Nguyen & Tran, 2021) | Dynamic (Nguyen & Tran, 2020) | Adap-Blend (Qi et al., 2023b) | Adap-Patch (Qi et al., 2023b) | AVG |
|---|---|---|---|---|---|---|---|---|---|---|
| STRIP (Gao et al., 2019) | cifar10 | 0.937 | 0.834 | 0.517 | 0.499 | 0.499 | 0.955 | 0.787 | 0.520 | 0.694 |
| | gtsrb | 0.955 | 0.772 | 0.670 | 0.500 | 0.500 | 0.971 | 0.917 | 0.577 | 0.733 |
| AC (Chen et al., 2018) | cifar10 | 0.999 | 0.992 | 1.000 | 0.500 | 0.500 | 0.958 | 0.958 | 1.000 | 0.863 |
| | gtsrb | 0.322 | 0.435 | 0.255 | 0.501 | 0.501 | 0.696 | 0.694 | 0.787 | 0.524 |
| Frequency (Zeng et al., 2021) | cifar10 | 1.000 | 0.936 | 1.000 | 0.999 | 0.999 | 0.969 | 0.896 | 0.902 | 0.963 |
| | gtsrb | 0.999 | 0.939 | 0.999 | 0.998 | 0.998 | 0.959 | 0.832 | 0.879 | 0.950 |
| SentiNet (Chou et al., 2018) | cifar10 | 0.949 | 0.463 | 0.949 | 0.502 | 0.502 | 0.949 | 0.470 | 0.947 | 0.716 |
| | gtsrb | 0.949 | 0.590 | 0.949 | 0.503 | 0.503 | 0.949 | 0.814 | 0.949 | 0.776 |
| CT (Qi et al., 2023c) | cifar10 | 0.9898 | 0.921 | 0.999 | 0.502 | 0.502 | 0.991 | 0.954 | 0.859 | 0.840 |
| | gtsrb | 0.967 | 0.978 | 0.999 | 0.504 | 0.504 | 0.955 | 0.983 | 0.861 | 0.844 |
| SS Tran et al. (2018) | cifar10 | 0.929 | 0.921 | 0.446 | 0.503 | 0.503 | 0.920 | 0.926 | 0.830 | 0.747 |
| | gtsrb | 0.808 | 0.722 | 0.800 | 0.502 | 0.502 | 0.800 | 0.722 | 0.680 | 0.692 |
| SCAn (Tang et al., 2021) | cifar10 | 0.985 | 0.983 | 0.986 | 0.498 | 0.498 | 0.991 | 0.815 | 0.819 | 0.822 |
| | gtsrb | 0.994 | 0.956 | 1.000 | 0.500 | 0.500 | 0.968 | 0.845 | 0.867 | 0.829 |
| SPECTRE (Hayase et al., 2021) | cifar10 | 0.895 | 0.765 | 0.931 | 0.545 | 0.545 | 0.841 | 0.5123 | 0.396 | 0.679 |
| | gtsrb | 0.911 | 0.599 | 0.800 | 0.502 | 0.502 | 0.567 | 0.615 | 0.626 | 0.640 |
| MM-BD (Wang et al., 2024) | cifar10 | 0.867 | 0.633 | 0.867 | 0.867 | 0.867 | 0.867 | 0.867 | 0.867 | 0.838 |
| | gtsrb | 0.567 | 0.633 | 0.500 | 0.633 | 0.767 | 0.567 | 0.833 | 0.833 | 0.667 |
| TED (Mo et al., 2024) | cifar10 | 0.642 | 0.485 | 0.503 | 0.411 | 0.676 | 0.433 | 0.526 | 0.664 | 0.543 |
| | gtsrb | 0.842 | 0.843 | 0.558 | 0.589 | 0.501 | 0.663 | 0.885 | 0.864 | 0.718 |
| BPROM (10%) | cifar10 | 1.000 | 1.000 | 1.000 | 1.000 | 1.000 | 1.000 | 1.000 | 1.000 | 1.000 |
| | gtsrb | 1.000 | 1.000 | 1.000 | 0.933 | 0.933 | 1.000 | 1.000 | 1.000 | 0.983 |

rate (ASR) > 0.98, shown in Table 14 of Section B.1). The meta-classifier, trained on Badnets-infected shadow models, classifies suspicious models under 9 attacks. Results from 30 clean and 30 backdoored suspicious models (Section 6.1) show BPROM outperforms all other defenses in average AUROC, even when the attack differs from the one used to train the meta-classifier.

BPROM achieves high AUROC using only 10% of the CIFAR-10 (GTSRB) test dataset as the reserved clean dataset $D_S$. Please see Table 23 in Section B.2 for BPROM (5%)'s and BPROM (1%)'s results. In contrast, baseline defenses' AUROC varies significantly across attacks and is heavily influenced by backdoor type. Defenses using activations or saliency maps fail against invisible backdoors spread throughout the image (Qi et al., 2023a), while perturbation and frequency-based methods cannot handle sample-specific or randomized triggers (Nguyen & Tran, 2021; 2020). Tables 17 and 18 in Section B.2 show that BPROM maintains high AUROC even with different architectures like MobileNetV2. We also evaluate BPROM on MobileViT and Swim Transformer, demonstrating its effectiveness across different architectures (see Section B.3 for details). We also tested feature-based backdoors like Refool (Yunfei Liu, 2020), BPP (Wang et al., 2022), and Poison Ink (Zhang et al., 2022), with results in Table 22 of Section B.2 showing perfect detection.

**Performance on Tiny-ImageNet and ImageNet.** In addition to the CIFAR-10 and GTSRB datasets, we also evaluated BPROM on the Tiny-ImageNet and ImageNet datasets. These larger datasets present greater challenges for backdoor detection due to the increased complexity of the images and the larger number of classes. Table 6 (Table 26 in Section D) shows the results on Tiny-ImageNet (ImageNet), comparing BPROM with several state-of-the-art defenses. In particular, for Tiny-ImageNet, BPROM achieves an average AUROC of 0.899 for ResNet18 and 0.912 for MobileNet, significantly outperforming other defenses.

**Training Time of BPROM.** BPROM's training time, while longer due to shadow model and meta-classifier training, remains practical for deployment given its accuracy and black-box nature. BPROM's training time with different shadow model counts and architectures (CIFAR-10 as $D_S$, STL-10 as $D_T$) is shown below. In particular, for ResNet18, BPROM's training time is 2.3, 4.8, and 9.5 hours if 10, 20, 40 shadow models are considered, respectively. For MobileNetV2, BPROM's training time is 1.2, 2.4, and 5.2 hours if 10, 20, 40 shadow models are considered, respectively. Reported times are averaged over five trials.

Table 6: AUROC of defenses on Tiny-ImageNet, using ResNet18 and MobileNetV2. AVG stands for the average AUROC. Green (red) cells denote values greater (lower) than 0.8.

| | | Badnets | Blend | Trojan | BPP | WaNet | Adap-Blend | Adap-Patch | AVG |
|---|---|---|---|---|---|---|---|---|---|
| STRIP (Gao et al., 2019) | ResNet18 | 0.938 | 0.905 | 0.440 | 0.500 | 0.500 | 0.914 | 0.925 | 0.732 |
| | MobileNetV2 | 0.936 | 0.935 | 0.940 | 0.500 | 0.500 | 0.838 | 0.830 | 0.783 |
| AC (Chen et al., 2018) | ResNet18 | 0.490 | 0.475 | 0.473 | 0.501 | 0.501 | 0.492 | 0.491 | 0.489 |
| | MobileNetV2 | 0.489 | 0.485 | 0.480 | 0.500 | 0.500 | 0.617 | 0.487 | 0.508 |
| SS Tran et al. (2018) | ResNet18 | 0.505 | 0.485 | 0.488 | 0.499 | 0.499 | 0.487 | 0.502 | 0.495 |
| | MobileNetV2 | 0.487 | 0.486 | 0.487 | 0.502 | 0.502 | 0.488 | 0.500 | 0.493 |
| SCAn (Tang et al., 2021) | ResNet18 | 0.987 | 0.987 | 0.994 | 0.502 | 0.502 | 0.741 | 0.788 | 0.786 |
| | MobileNetV2 | 0.982 | 0.987 | 0.986 | 0.502 | 0.502 | 0.888 | 0.882 | 0.818 |
| CT (Qi et al., 2023c) | ResNet18 | 0.945 | 0.936 | 0.882 | 0.501 | 0.501 | 0.778 | 0.776 | 0.760 |
| | MobileNetV2 | 0.889 | 0.864 | 0.915 | 0.500 | 0.500 | 0.823 | 0.818 | 0.758 |
| SCALE-UP (Guo et al., 2023) | ResNet18 | 0.742 | 0.724 | 0.515 | 1.000 | 1.000 | 0.515 | 0.606 | 0.729 |
| | MobileNetV2 | 0.651 | 0.548 | 0.510 | 0.980 | 0.980 | 0.510 | 0.717 | 0.699 |
| CD (Huang et al., 2023) | ResNet18 | 0.918 | 0.954 | 0.961 | 0.628 | 0.628 | 0.542 | 0.647 | 0.754 |
| | MobileNetV2 | 0.904 | 0.985 | 0.997 | 0.514 | 0.514 | 0.591 | 0.933 | 0.805 |
| MM-BD (Wang et al., 2024) | ResNet18 | 0.800 | 0.567 | 0.467 | 0.967 | 0.467 | 0.867 | 0.867 | 0.715 |
| | MobileNetV2 | 0.633 | 0.500 | 0.467 | 1.000 | 0.700 | 0.633 | 0.767 | 0.671 |
| BPROM (10%) | ResNet18 | 1.000 | 0.984 | 0.900 | 1.000 | 1.000 | 0.966 | 1.000 | 0.979 |
| | MobileNetV2 | 1.000 | 0.978 | 0.966 | 1.000 | 1.000 | 1.000 | 1.000 | 0.992 |

## 6.3 HYPERPARAMETER STUDY

We conduct hyperparameter studies to analyze key factors affecting BPROM's effectiveness.

**Impact of Number of Shadow Models.** Table 7 shows AUROC as we vary the number of shadow models used to train the backdoor classifier. In the table, "2 (1+1)" means one clean and one backdoor shadow model. The F1 score increases rapidly with more shadow models but plateaus after about 20 models. This indicates that approximately 20 shadow models are sufficient for effective training, with minimal AUROC improvement beyond this number.

**Impact of Trigger Size and Poison Rate.** We analyze how detection performance (AUROC) changes with varying trigger size and poison rate. The settings in Tables 8 and 9 match those in Tables 3 and 4, which show the prompted model accuracy for different trigger sizes and poison rates. Tables 8 and 9 show both attack success rate (ASR) and AUROC for CIFAR-10 models as trigger size and poison rate vary.

We observe two key points: 1) ASR increases with larger trigger sizes and poison rates, indicating stronger backdoor attacks. 2) Despite stronger attacks, our detection method's AUROC remains stable, with minor fluctuations. GTSRB results show similar trends: as trigger size increases from 4×4 to 16×16, ASR rises from 26% to 99%, while AUROC stays between 0.98 and 1.00. This demonstrates that our back-

Table 7: AUROC relative to the number of shadow models in meta-classifier training.

| | CIFAR-10 | | GTSRB | |
|---|---|---|---|---|
| # Shadow Model | Blend | Adap-Blend | Blend | Adap-Blend |
| 2 (1+1) | 0.667 | 0.938 | 0.789 | 0.967 |
| 10 (5+5) | 0.874 | 0.985 | 0.854 | 0.989 |
| 20 (10+10) | 1.000 | 1.000 | 1.000 | 1.000 |
| 40 (20+20) | 1.000 | 1.000 | 1.000 | 1.000 |

Table 8: ASR and AUROC for Blend and Adap-Blend attacks across different trigger sizes.

| | Trigger Size | Blend | | Adap-Blend | |
|---|---|---|---|---|---|
| | | ASR | AUROC | ASR | AUROC |
| CIFAR-10 | (4*4) | 0.269 | 1.000 | 0.016 | 1.000 |
| | (8*8) | 0.974 | 1.000 | 0.049 | 1.000 |
| | (16*16) | 0.994 | 1.000 | 0.963 | 1.000 |
| GTSRB | (4*4) | 0.842 | 1.000 | 0.027 | 1.000 |
| | (8*8) | 0.994 | 1.000 | 0.194 | 1.000 |
| | (16*16) | 0.994 | 1.000 | 0.997 | 1.000 |

Table 9: ASR and AUROC for Blend and Adap-Blend attacks at different poison rates.

| | Poison Rate | Blend | | Adap-Blend | |
|---|---|---|---|---|---|
| | | ASR | AUROC | ASR | AUROC |
| CIFAR-10 | 5% | 0.996 | 0.607 | 0.998 | 0.607 |
| | 10% | 0.990 | 0.933 | 0.998 | 0.909 |
| | 20% | 0.998 | 1.000 | 1.000 | 1.000 |
| GTSRB | 5% | 0.998 | 1.000 | 1.000 | 1.000 |
| | 10% | 0.998 | 1.000 | 1.000 | 1.000 |
| | 20% | 0.991 | 1.000 | 1.000 | 1.000 |

door detection technique remains reliable even as attacks strengthen, highlighting its robustness against varying attack strengths.

**Structural Differences between Shadow and Suspicious Models.** We analyze the impact of using different architectures for shadow and suspicious models on BPROM's performance. Table 10 shows AUROC results with MobileNetV2 as the suspicious model and ResNet18 as the shadow model, indicating that BPROM's detection effectiveness remains robust despite structural differences.

**Impact of External Dataset.** We ran additional experiments with $D_S$ as CIFAR-10/GTSRB and $D_T$ changed to SVHN. Results in Tables 19 and 20 of Section B.2 show consistent detection performance.

Table 10: F1 score and AUROC of BPROM when the suspicious model is MobileNetV2 and the shadow model is ResNet18.

|  | WaNet | Adap-Blend | Adap-Patch | AVG |
|---|---|---|---|---|
| F1 | 1.000 | 1.000 | 1.000 | 1.000 |
| AUROC | 1.000 | 1.000 | 1.000 | 1.000 |

**Impact of the Inconsistency between Numbers of classes in $D_S$ and $D_T$.** In previous experiments, we used CIFAR-10 and GTSRB as $D_S$ and STL-10 as $D_T$, maintaining class consistency between $D_S$ and $D_T$. We also ran experiments with $D_T$ as STL-10 and $D_S$ as CIFAR-100. The results in Table 21 of Section B.2 still show consistent detection performance.

## 6.4 ADAPTIVE ATTACK

To evaluate BPROM's robustness against adaptive attacks, we followed the experimental setup described in Guo et al. (2023) (Section 5.3.2), focusing on BadNets attacks on CIFAR-10. It remains unknown how an attacker adds a regularization term to reduce class subspace inconsistency. We examine two candidate adaptive attacks below.

Table 11: Adaptive attacks with low poison rate.

| Poison Rate | AUROC | ASR | Poison Rate | AUROC | ASR |
|---|---|---|---|---|---|
| 0.2% | 1 | 0.709 | 2% | 1 | 1 |
| 0.5% | 1 | 0.838 | 5% | 1 | 1 |
| 1% | 1 | 1 | 10% | 1 | 1 |

First, as shown in Qi et al. (2023b), the backdoor with a very low poison rate can act as an adaptive attack. Table 11 presents the AUROC and ASR of BPROM at various poison rates. These results show that BPROM maintains perfect detection (AUROC = 1) even at extremely low poison rates, demonstrating its effectiveness against stealthy adaptive attacks. Our observed ASR values for BadNets at 0.2% and 0.5% poison rates align with those reported in Figure 7b of Guo et al. (2023), validating the correctness of our implementation.

Clean-label backdoors, like SIG (Barni et al., 2019) and LC (Turner et al., 2019) can also be regarded as a different adaptive attack. These attacks do not modify labels and only poison a portion of the training images, potentially preserving class subspaces and hindering BPROM's

Table 12: Adaptive attacks with clean labels.

| Dataset | SIG | LC |
|---|---|---|
| CIFAR-10 | 1.00 | 0.95 |
| GTSRB | 0.83 | 0.78 |

detection based on class subspace inconsistency. BPROM. Table 12 shows BPROM's performance on SIG and LC. While not perfect, BPROM still achieves decent AUROC, indicating its resilience even against these challenging attacks.

## 7 CONCLUSION AND LIMITATION

We present BPROM as a novel VP-based black-box model-level backdoor detection method. BPROM relies on class subspace inconsistency, where the prompted model's accuracy degrades if the source model is backdoored. This inconsistency is common in various backdoor attacks due to feature space distortion from the poisoned dataset. Our experiments show BPROM effectively detects all-to-one backdoors. However, it struggles with all-to-all backdoors, as their feature space distortion is more controllable by the attacker. Addressing this limitation is left for future work.

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

# APPENDIX OF BPROM: BLACK-BOX MODEL-LEVEL BACKDOOR DETECTION VIA VISUAL PROMPTING

This appendix provides additional details and experimental results supporting our main findings. Section A details the implementation and configurations of the experiments. Section B presents BPROM's evaluation on different model architectures, datasets, and attack settings, including analyses of label mapping, class number inconsistency, and feature-based backdoors. Section C provides additional visualizations of class subspace inconsistency to further illustrate our method.

## A IMPLEMENTATION DETAILS

We provide details on the configurations of the experiments used to evaluate BPROM and other defenses.

### A.1 ATTACK CONFIGURATIONS

The configurations of the baseline attacks used in our experiments are summarized in Table 13. For each attack, we specify parameters related to the backdoor trigger insertion, including poison rate and cover rate.

- Poison rate: The proportion of training data with the trigger pattern. A higher poison rate increases the attacker's influence on the model's behavior but also raises the detection risk.
- Cover rate: The proportion of data with the trigger pattern that shares the original label. A higher cover rate makes the trigger pattern more stealthy and consistent with the original data distribution but weakens the attack.

All attacks are implemented using the default settings in the Backdoor Toolbox[4]; refer to the code repository for more details.

Table 13: Configurations of baseline attacks

| Attacks | CIFAR-10 | GTSRB |
|---|---|---|
| BadNets Gu et al. (2017) | Poison Rate: 0.3% | Poison Rate: 1.0% |
| Blend Chen et al. (2017) | Poison Rate: 0.3% | Poison Rate: 1.0% |
| Trojan Liu et al. (2018b) | Poison Rate: 0.3% | Poison Rate: 1.0% |
| WaNet Nguyen & Tran (2021) | Poison Rate: 5.0% Cover Rate: 10.0% | Poison Rate: 5.0% Cover Rate: 10.0% |
| Dynamic Nguyen & Tran (2020) | Poison Rate: 0.3% | Poison Rate: 0.3% |
| Adap-Blend Qi et al. (2023b) | Poison Rate: 0.3% Cover Rate: 0.6% | Poison Rate: 0.5% Cover Rate: 1.0% |
| Adap-Patch Qi et al. (2023b) | Poison Rate: 0.3% Cover Rate: 0.3% | Poison Rate: 0.3% Cover Rate: 0.6% |

### A.2 DEFENSE CONFIGURATIONS

The important settings used for baseline defenses in our evaluations are summarized below:

- **STRIP** (Gao et al., 2019): Number of superimposing images = 10; defense false positive rate budget = 10%.
- **AC** (Chen et al., 2018): Cluster threshold = 35% of class size.
- **Frequency** (Zeng et al., 2021): Predicts samples as poisoned or clean using a pretrained binary classifier.
- **SentiNet** (Chou et al., 2018): FPR = 5%, number of high activation pixels = top 15%.
- **CT** (Qi et al., 2023c): Confusion iterations = 6000; confusion factor = 20.
- **SS** (Tran et al., 2018): Number of removed samples = $\min(1.5 \times |D_{poison}|/|D|, 0.5 \times$ class size).

---

[4]https://github.com/vtu81/backdoor-toolbox

- **SCAn** (Tang et al., 2021): Threshold for abnormal score = 0.5.
- **SPECTRE** (Hayase et al., 2021): Number of removed samples = $\min(1.5 * |D_{poison}|/|D|, 0.5 \times$ class size$)$ from top 50% suspicious classes.

# B  EVALUATIONS ON DIFFERENT ARCHITECTURES AND DATASETS

To evaluate the effectiveness of BPROM on different architectures, we conducted experiments using ResNet (He et al., 2015) and MobileNetV2 (Sandler et al., 2018) as backbone models. . The models are trained on the CIFAR-10 (Krizhevsky, 2009) and GTSRB (Stallkamp et al., 2011) datasets, attacked with 9 different backdoor attacks, and then defended with state-of-the-art methods.

## B.1  ACCURACY AND ATTACK SUCCESS RATE

We report the clean accuracy (ACC) of the infected models on benign test samples without triggers and the attack success rate (ASR), which indicates the percentage of Trojan inputs successfully predicted as the attacker-specified target class. The results are shown in Table 14 for ResNet18 and Table 15 for MobileNetV2.

Table 14: Accuracy and ASR on ResNet18.

| | | Badnets Gu et al. (2017) | Blend Chen et al. (2017) | Trojan Liu et al. (2018b) | WaNet Nguyen & Tran (2021) | Dynamic Nguyen & Tran (2020) | Adap-Blend Qi et al. (2023b) | Adap-Patch Qi et al. (2023b) | Clean |
|---|---|---|---|---|---|---|---|---|---|
| CIFAR-10 | ACC | 0.936 | 0.934 | 0.939 | 0.926 | 0.941 | 0.933 | 0.936 | 0.937 |
| | ASR | 1.000 | 0.998 | 1.000 | 0.987 | 0.998 | 0.998 | 1.000 | - |
| GTSRB | ACC | 0.968 | 0.968 | 0.972 | 0.952 | 0.971 | 0.971 | 0.974 | 0.976 |
| | ASR | 1.000 | 0.996 | 1.000 | 0.986 | 1.000 | 0.995 | 0.982 | - |

Table 15: Accuracy and ASR on MobileNetV2.

| | | Badnets Gu et al. (2017) | Blend Chen et al. (2017) | Trojan Liu et al. (2018b) | WaNet Nguyen & Tran (2021) | Dynamic Nguyen & Tran (2020) | Adap-Blend Qi et al. (2023b) | Adap-Patch Qi et al. (2023b) | Clean |
|---|---|---|---|---|---|---|---|---|---|
| CIFAR-10 | ACC | 0.905 | 0.906 | 0.901 | 0.907 | 0.905 | 0.898 | 0.902 | 0.906 |
| | ASR | 1.000 | 0.994 | 1.000 | 0.990 | 1.000 | 1.000 | 1.000 | - |
| GTSRB | ACC | 0.935 | 0.927 | 0.938 | 0.905 | 0.922 | 0.921 | 0.937 | 0.931 |
| | ASR | 1.000 | 0.994 | 1.000 | 0.991 | 1.000 | 1.000 | 1.000 | - |

The results presented in Table 14 and Table 15 reveal that despite maintaining high clean accuracy, both models exhibit very high attack success rates (>98%) across various attacks when triggers are present. This suggests that the backdoors effectively induce misclassification towards the target label. With the effectiveness of the backdoor attacks established, the subsequent evaluation involves assessing the performance of BPROM and other state-of-the-art defense methods in detecting these compromised models.

## B.2  AUROC AND F1 SCORE

We evaluate defense methods in detecting backdoor attacks using AUROC and F1 score metrics. Experiments are conducted on CIFAR-10 and GTSRB datasets using ResNet18 and MobileNetV2 architectures to assess and compare detection effectiveness across different model designs. This allows for determining the robustness and architecture-agnostic capability of techniques.

**Experiments on ResNet18.**  From the AUROC results in Table 25 and F1 scores in Table 16 of defenses evaluated on the ResNet18 model, we observe that BPROM demonstrates competitive or superior detection performance over defenses for the majority of attacks. It also significantly elevates the average AUROC and F1 score over the strongest baselines. Although it exhibits relatively lower scores on two attacks, BPROM still demonstrates detection capability on par with or better than other methods.

Table 16: F1 scores of defense methods against backdoor attacks in CIFAR-10 and GTSRB. AVG stands for the average F1 score.

| | | Badnets Gu et al. (2017) | Blend Chen et al. (2017) | Trojan Liu et al. (2018b) | WaNet Nguyen & Tran (2021) | Dynamic Nguyen & Tran (2020) | Adap-Blend Qi et al. (2023b) | Adap-Patch Qi et al. (2023b) | AVG |
|---|---|---|---|---|---|---|---|---|---|
| STRIP Gao et al. (2019) | cifar10 | 0.952 | 0.466 | 0.951 | 0.471 | 0.951 | 0.848 | 0.009 | 0.664 |
| | gtsrb | 0.952 | 0.851 | 0.924 | 0.489 | 0.952 | 0.937 | 0.052 | 0.737 |
| AC Chen et al. (2018) | cifar10 | 1.000 | 0.946 | 1.000 | 0.883 | 0.978 | 1.000 | 0.000 | 0.830 |
| | gtsrb | 0.000 | 0.000 | 0.000 | 0.000 | 0.000 | 0.000 | 0.000 | 0.000 |
| Frequency Zeng et al. (2021) | cifar10 | 1.000 | 0.921 | 1.000 | 0.141 | 0.981 | 0.921 | 0.784 | 0.821 |
| | gtsrb | 0.854 | 0.812 | 0.854 | 0.361 | 0.792 | 0.814 | 0.679 | 0.738 |
| SentiNet Chou et al. (2018) | cifar10 | 0.952 | 0.114 | 0.291 | 0.170 | 0.596 | 0.121 | 0.957 | 0.457 |
| | gtsrb | 0.952 | 0.434 | 0.952 | 0.484 | 0.721 | 0.792 | 0.975 | 0.759 |
| CT Qi et al. (2023c) | cifar10 | 0.470 | 0.630 | 0.949 | 0.682 | 0.664 | 0.908 | 0.965 | 0.753 |
| | gtsrb | 0.747 | 0.654 | 0.576 | 0.962 | 0.916 | 0.892 | 0.965 | 0.816 |
| SS Tran et al. (2018) | cifar10 | 0.979 | 0.936 | 0.294 | 0.741 | 0.789 | 0.661 | 0.0208 | 0.632 |
| | gtsrb | 0.829 | 0.807 | 0.965 | 0.530 | 0.875 | 0.538 | 0.681 | 0.746 |
| SCAn Tang et al. (2021) | cifar10 | 0.993 | 0.964 | 0.991 | 0.935 | 0.979 | 0.000 | 0.000 | 0.695 |
| | gtsrb | 0.990 | 0.966 | 0.999 | 0.956 | 0.874 | 0.000 | 0.000 | 0.684 |
| SPECTRE Hayase et al. (2021) | cifar10 | 0.990 | 0.990 | 0.991 | 0.839 | 0.991 | 0.938 | 0.865 | 0.943 |
| | gtsrb | 0.957 | 0.954 | 0.968 | 0.000 | 0.976 | 0.000 | 0.000 | 0.551 |
| BPROM (10%) | cifar10 | 1.000 | 1.000 | 1.000 | 1.000 | 1.000 | 1.000 | 1.000 | **1.000** |
| | gtsrb | 1.000 | 1.000 | 1.000 | 1.000 | 1.000 | 1.000 | 1.000 | **1.000** |
| BPROM (5%) | cifar10 | 1.000 | 1.000 | 1.000 | 1.000 | 1.000 | 1.000 | 1.000 | 1.000 |
| | gtsrb | 1.000 | 0.965 | 1.000 | 1.000 | 1.000 | 1.000 | 1.000 | 0.995 |
| BPROM (1%) | cifar10 | 1.000 | 1.000 | 1.000 | 1.000 | 1.000 | 1.000 | 1.000 | 1.000 |
| | gtsrb | 1.000 | 0.782 | 1.000 | 1.000 | 1.000 | 1.000 | 1.000 | 0.969 |

**Experiments on MobileNetV2.** We further evaluate the effectiveness of backdoor detection methods when using the MobileNetV2 architecture, which utilizes depth-separable convolutions to build a lightweight model. This represents a different design choice than ResNet, which uses residual connections to train deeper models. As shown in Table 17 and Table 18, we observe consistently outstanding detection effectiveness of BPROM over defenses.

Table 17: AUROC of defenses on MobileNetV2 under backdoor attacks on CIFAR-10 and GTSRB. AVG stands for the average AUROC.

| | | Badnets Gu et al. (2017) | Blend Chen et al. (2017) | Trojan Liu et al. (2018b) | WaNet Nguyen & Tran (2021) | Dynamic Nguyen & Tran (2020) | Adap-Blend Qi et al. (2023b) | Adap-Patch Qi et al. (2023b) | AVG |
|---|---|---|---|---|---|---|---|---|---|
| STRIP Gao et al. (2019) | cifar10 | 0.739 | 0.833 | 0.957 | 0.473 | 0.987 | 0.987 | 0.952 | 0.847 |
| | gtsrb | 0.798 | 0.873 | 0.745 | 0.489 | 0.998 | 0.981 | 0.999 | 0.840 |
| AC Chen et al. (2018) | cifar10 | 0.364 | 0.398 | 0.996 | 0.889 | 0.425 | 0.390 | 1.000 | 0.637 |
| | gtsrb | 0.225 | 0.309 | 0.263 | 0.355 | 0.288 | 0.576 | 0.263 | 0.326 |
| Frequency Zeng et al. (2021) | cifar10 | 1.000 | 0.996 | 1.000 | 0.999 | 0.970 | 0.996 | 1.000 | 0.994 |
| | gtsrb | 1.000 | 0.973 | 1.000 | 0.777 | 0.960 | 0.973 | 1.000 | 0.955 |
| CT Qi et al. (2023c) | cifar10 | 0.999 | 0.996 | 0.999 | 0.943 | 0.985 | 0.992 | 0.802 | 0.959 |
| | gtsrb | 0.993 | 0.984 | 0.998 | 0.852 | 0.985 | 0.985 | 0.616 | 0.916 |
| SS Tran et al. (2018) | cifar10 | 0.439 | 0.428 | 0.375 | 0.381 | 0.426 | 0.377 | 0.442 | 0.410 |
| | gtsrb | 0.492 | 0.492 | 0.492 | 0.492 | 0.487 | 0.492 | 0.487 | 0.491 |
| SCAn Tang et al. (2021) | cifar10 | 0.991 | 0.921 | 0.953 | 0.926 | 0.988 | 0.981 | 0.926 | 0.955 |
| | gtsrb | 0.999 | 0.979 | 0.969 | 0.952 | 0.986 | 0.969 | 0.976 | 0.976 |
| SPECTRE Hayase et al. (2021) | cifar10 | 0.857 | 0.376 | 0.876 | 0.534 | 0.897 | 0.510 | 0.376 | 0.632 |
| | gtsrb | 0.911 | 0.699 | 0.797 | 0.597 | 0.595 | 0.617 | 0.581 | 0.685 |
| BPROM (10%) | cifar10 | 1.000 | 1.000 | 1.000 | 1.000 | 1.000 | 1.000 | 1.000 | **1.000** |
| | gtsrb | 1.000 | 0.999 | 1.000 | 1.000 | 1.000 | 1.000 | 1.000 | **1.000** |

The consistent behavior shows that the effectiveness of BPROM in detecting backdoors is preserved irrespective of model complexity and design choices.

Table 18: F1 score of defenses on MobileNetV2 under backdoor attacks on CIFAR-10 and GTSRB. AVG stands for the average F1 score.

| | | Badnets Gu et al. (2017) | Blend Chen et al. (2017) | Trojan Liu et al. (2018b) | WaNet Nguyen & Tran (2021) | Dynamic Nguyen & Tran (2020) | Adap-Blend Qi et al. (2023b) | Adap-Patch Qi et al. (2023b) | AVG |
|---|---|---|---|---|---|---|---|---|---|
| STRIP Gao et al. (2019) | cifar10 | 0.552 | 0.673 | 0.916 | 0.122 | 0.939 | 0.943 | 0.999 | 0.735 |
| | gtsrb | 0.513 | 0.800 | 0.442 | 0.148 | 0.954 | 0.937 | 0.955 | 0.678 |
| AC Chen et al. (2018) | cifar10 | 0.000 | 0.000 | 0.998 | 0.942 | 0.000 | 0.000 | 1.000 | 0.420 |
| | gtsrb | 0.000 | 0.000 | 0.000 | 0.000 | 0.000 | 0.000 | 0.000 | 0.000 |
| Frequency Zeng et al. (2021) | cifar10 | 0.983 | 0.922 | 0.984 | 0.981 | 0.139 | 0.92 | 0.983 | 0.844 |
| | gtsrb | 0.865 | 0.825 | 0.864 | 0.392 | 0.804 | 0.826 | 0.865 | 0.777 |
| CT Qi et al. (2023c) | cifar10 | 0.998 | 0.988 | 0.999 | 0.936 | 0.981 | 0.992 | 0.753 | 0.950 |
| | gtsrb | 0.976 | 0.967 | 0.992 | 0.810 | 0.960 | 0.968 | 0.378 | 0.864 |
| SS Tran et al. (2018) | cifar10 | 0.230 | 0.222 | 0.141 | 0.176 | 0.218 | 0.173 | 0.232 | 0.199 |
| | gtsrb | 0.278 | 0.279 | 0.278 | 0.278 | 0.272 | 0.278 | 0.278 | 0.277 |
| SCAn Tang et al. (2021) | cifar10 | 0.938 | 0.908 | 0.997 | 0.920 | 0.987 | 0.981 | 0.974 | 0.958 |
| | gtsrb | 0.642 | 0.769 | 1.000 | 0.818 | 0.968 | 0.450 | 0.335 | 0.712 |
| SPECTRE Hayase et al. (2021) | cifar10 | 0.774 | 0.676 | 0.125 | 0.282 | 0.775 | 0.252 | 0.169 | 0.436 |
| | gtsrb | 0.897 | 0.401 | 0.748 | 0.358 | 0.356 | 0.388 | 0.338 | 0.498 |
| BPROM (10%) | cifar10 | 1.000 | 1.000 | 1.000 | 0.967 | 1.000 | 1.000 | 1.000 | **0.995** |
| | gtsrb | 1.000 | 0.965 | 1.000 | 1.000 | 0.965 | 0.965 | 1.000 | **0.985** |

**Experiments on extra external dataset.** We ran extra experiments, where $D_S$ is kept as CIFAR-10/GTSRB, but $D_T$ is changed to SVHN. Table 19 shows the results when $D_S$ is GTSRB and Table 20 shows the results when $D_S$ is CIFAR-10. Both results demonstrate consistent detection performance of BPROM even when using a different external dataset $D_T$. This indicates that the choice of external dataset does not significantly impact BPROM's effectiveness.

Table 19: $D_T$ is changed to SVHN, $D_S$ is kept as GTSRB.

| | Badnets (Gu et al., 2017) | Blend (Chen et al., 2017) | Trojan (Liu et al., 2018b) | WaNet (Nguyen & Tran, 2021) | Dynamic (Nguyen & Tran, 2020) | Adap-Blend (Qi et al., 2023b) | Adap-Patch (Qi et al., 2023b) | AVG |
|---|---|---|---|---|---|---|---|---|
| F1 | 0.882 | 1.000 | 1.000 | 0.667 | 1.000 | 0.937 | 1.000 | 0.927 |
| AUROC | 0.867 | 1.000 | 1.000 | 0.500 | 1.000 | 0.933 | 1.000 | 0.9001 |

Table 20: $D_T$ is changed to SVHN, $D_S$ is kept as CIFAR-10.

| | Badnets (Gu et al., 2017) | Blend (Chen et al., 2017) | Trojan (Liu et al., 2018b) | WaNet (Nguyen & Tran, 2021) | Dynamic (Nguyen & Tran, 2020) | Adap-Blend (Qi et al., 2023b) | Adap-Patch (Qi et al., 2023b) | AVG |
|---|---|---|---|---|---|---|---|---|
| F1 | 1.000 | 1.000 | 1.000 | 0.967 | 1.000 | 1.000 | 1.000 | 0.995 |
| AUROC | 1.000 | 1.000 | 1.000 | 0.967 | 1.000 | 1.000 | 1.000 | 0.995 |

**Experiments on CIFAR-100.** To investigate the impact of inconsistency between the numbers of classes in $D_S$ and $D_T$, we conducted experiments using CIFAR-100 as $D_S$ and STL-10 as $D_T$. Table 21 shows that BPROM achieves high AUROC and F1 scores across various backdoor attacks, demonstrating its robustness even when there is a significant mismatch in the number of classes (100 classes in $D_S$ vs. 10 classes in $D_T$). This suggests that BPROM is capable of handling scenarios where the source and target domains have different numbers of classes, making it a versatile detection method.

**Experiments on feature-based backdoors.** We further evaluated BPROM's performance on feature-based backdoors, which manipulate the model's feature representations instead of directly modifying

Table 21: AUROC of defenses on ResNet18 under backdoor attacks on CIFAR-100. AVG stands for the average AUROC.

| | Badnets (Gu et al., 2017) | Blend (Chen et al., 2017) | Trojan (Liu et al., 2018b) | WaNet (Nguyen & Tran, 2021) | Adap-Blend (Qi et al., 2023b) | Adap-Patch (Qi et al., 2023b) | AVG |
|---|---|---|---|---|---|---|---|
| STRIP (Gao et al., 2019) | 0.876 | 0.732 | 0.762 | 0.135 | 0.941 | 0.964 | 0.729 |
| AC (Chen et al., 2018) | 0.000 | 0.000 | 0.000 | 0.000 | 0.998 | 1.000 | 0.333 |
| Frequency (Zeng et al., 2021) | 0.986 | 0.896 | 0.895 | 0.883 | 0.865 | 0.985 | 0.926 |
| SentiNet Chou et al. (2018) | 0.952 | 0.047 | 0.952 | 0.115 | 0.240 | 0.952 | 0.551 |
| SS (Tran et al., 2018) | 0.661 | 0.005 | 0.661 | 0.673 | 0.633 | 0.672 | 0.551 |
| SCAn(Tang et al., 2021) | 0.992 | 0.980 | 0.982 | 0.612 | 0.877 | 0.380 | 0.804 |
| BPROM(10%) | 1.000 | 1.000 | 1.000 | 1.000 | 1.000 | 1.000 | **1.000** |

input images. Table 22 presents the results of BPROM on three feature-based backdoor methods: Refool (Yunfei Liu, 2020), BPP (Wang et al., 2022), and Poison Ink (Zhang et al., 2022), using the same configuration as previous experiments. The high F1 scores and AUROC values indicate that BPROM effectively detects these feature-based backdoors, demonstrating its versatility in handling diverse backdoor attack strategies.

Table 22: Feature-based backdoors like Refool, BPP, Poison Ink.

| Attack | Dataset | F1 Score | AUROC |
|---|---|---|---|
| Refool (Yunfei Liu, 2020) | CIFAR-10 | 1.000 | 1.000 |
| BPP (Wang et al., 2022) | CIFAR-10 | 1.000 | 1.000 |
| Poison Ink (Zhang et al., 2022) | CIFAR-10 | 1.000 | 1.000 |

**Impact of Reserved Clean Dataset Size.** We analyze the impact of the reserved clean dataset size ($D_S$) on BPROM's performance. As shown in Table 23, BPROM maintains high AUROC across different $D_S$ sizes (1%, 5%, and 10% of the CIFAR-10 and GTSRB test sets). Even with a limited $D_S$ (1%), BPROM achieves competitive performance, demonstrating its efficiency in leveraging small amounts of clean data. This robustness to $D_S$ size makes BPROM practical for real-world scenarios where clean data might be scarce.

### B.3 BPROM PERFORMANCE ON MOBILEVIT AND SWIM TRANSFORMER

To demonstrate BPROM's architecture-agnostic nature, we evaluated its performance on MobileViT and Swim Transformer, models combining CNN and transformer components. Tables 24 and 25 present the AUROC scores on CIFAR-10 and GTSRB across various backdoor attacks. The results show that BPROM maintains competitive performance on both MobileViT and Swim Transformer, indicating its effectiveness is not limited to ResNet-based architectures. The average AUROC is calculated for each defense and dataset.

## C ANOTHER VISUALIZATION OF CLASS SUBSPACE INCONSISTENCY

Figure 5a illustrates, using principal component analysis (PCA), 30 suspicious models (15 clean and 15 backdoor) trained on the complete CIFAR-10 dataset, along with 40 shadow models (20 clean-shadow and 20 backdoor-shadow) trained on 10% of the CIFAR-10 test set. All models are based on ResNet18, with the Trojan method Liu et al. (2018b) employed as the backdoor technique. Subsequently, a random forest-based meta-model (binary classifier) with 50 estimators is trained on the confidence vectors produced by the 40 shadow models. A distinct separation between clean (green

Table 23: Detailed AUROC of BPROM with varying sizes of the reserved clean dataset ($D_S$).

| | | Badnets (Gu et al., 2017) | Blend (Chen et al., 2017) | Trojan (Liu et al., 2018b) | WaNet (Nguyen & Tran, 2021) | Dynamic (Nguyen & Tran, 2020) | Adap-Blend (Qi et al., 2023b) | Adap-Patch (Qi et al., 2023b) | AVG |
|---|---|---|---|---|---|---|---|---|---|
| BPROM (10%) | cifar10 | 1.000 | 1.000 | 1.000 | 1.000 | 1.000 | 1.000 | 1.000 | **1.000** |
| | gtsrb | 1.000 | 1.000 | 1.000 | 1.000 | 1.000 | 1.000 | 1.000 | **1.000** |
| BPROM (5%) | cifar10 | 1.000 | 1.000 | 1.000 | 1.000 | 1.000 | 1.000 | 1.000 | **1.000** |
| | gtsrb | 1.000 | 1.000 | 1.000 | 1.000 | 1.000 | 1.000 | 1.000 | **1.000** |
| BPROM (1%) | cifar10 | 1.000 | 1.000 | 1.000 | 1.000 | 1.000 | 1.000 | 1.000 | **1.000** |
| | gtsrb | 1.000 | 1.000 | 1.000 | 1.000 | 1.000 | 1.000 | 1.000 | **1.000** |

Table 24: AUROC on MobileViT for CIFAR-10 and GTSRB. AVG stands for the average AUROC. Green (red) cells denote values greater (lower) than 0.8.

| | | Badnets (Gu et al., 2017) | Blend (Chen et al., 2017) | Trojan (Liu et al., 2018b) | WaNet (Nguyen & Tran, 2021) | Dynamic (Nguyen & Tran, 2020) | Adap-Blend (Qi et al., 2023b) | Adap-Patch (Qi et al., 2023b) | AVG |
|---|---|---|---|---|---|---|---|---|---|
| STRIP (Gao et al., 2019) | cifar10 | 0.4974 | 0.7775 | 0.9495 | 0.4705 | 0.9555 | 0.9497 | 0.9501 | 0.7929 |
| | gtsrb | 0.9140 | 0.900 | 0.9497 | 0.4855 | 0.9501 | 0.9447 | 0.9501 | 0.8706 |
| AC (Chen et al., 2018) | cifar10 | 0.4738 | 0.7745 | 1.0000 | 0.4252 | 0.8334 | 0.9930 | 0.8996 | 0.7714 |
| | gtsrb | 0.2198 | 0.2591 | 1.0000 | 0.312 | 0.6702 | 0.9999 | 1.0000 | 0.6373 |
| Frequency (Zeng et al., 2021) | cifar10 | 1.000 | 0.9962 | 1.000 | 0.969 | 0.9999 | 0.9961 | 0.9643 | 0.9893 |
| | gtsrb | 0.9994 | 0.9732 | 0.9993 | 0.7782 | 0.9601 | 0.8731 | 0.8993 | 0.9261 |
| CT (Qi et al., 2023c) | cifar10 | 0.9941 | 0.9379 | 0.9843 | 0.7475 | 0.9892 | 0.9439 | 0.8815 | 0.9255 |
| | gtsrb | 0.9727 | 0.9718 | 0.9744 | 0.9677 | 0.5652 | 0.9684 | 0.9289 | 0.9070 |
| SS Tran et al. (2018) | cifar10 | 0.5002 | 0.3902 | 0.3745 | 0.5145 | 0.6154 | 0.3801 | 0.3977 | 0.4532 |
| | gtsrb | 0.4925 | 0.4987 | 0.4925 | 0.4925 | 0.4966 | 0.4961 | 0.4929 | 0.4945 |
| SCAn (Tang et al., 2021) | cifar10 | 0.5000 | 0.5000 | 0.9999 | 0.6578 | 0.5652 | 0.5000 | 0.5000 | 0.6104 |
| | gtsrb | 0.4771 | 0.5481 | 0.4747 | 0.8439 | 0.5000 | 0.8758 | 0.6902 | 0.6300 |
| SPECTRE (Hayase et al., 2021) | cifar10 | 0.4752 | 0.4752 | 0.5961 | 0.4754 | 0.4754 | 0.5015 | 0.4752 | 0.4963 |
| | gtsrb | 0.7383 | 0.4995 | 0.7383 | 0.4995 | 0.4995 | 0.4995 | 0.4991 | 0.5677 |
| BPROM (10%) | cifar10 | 1.0000 | 1.0000 | 0.9667 | 1.0000 | 1.0000 | 1.0000 | 1.0000 | 0.9952 |
| | gtsrb | 1.0000 | 1.0000 | 1.0000 | 1.0000 | 0.9655 | 0.9655 | 1.0000 | 0.9901 |

dots) and backdoor models (blue dots) is evident after VP, attributed to class subspace inconsistency. The same meta-model is also used to classify clean (green dots) and Adap-Blend-infected models (red dots) Qi et al. (2023b). A similar pattern is observable in Figure 5b.

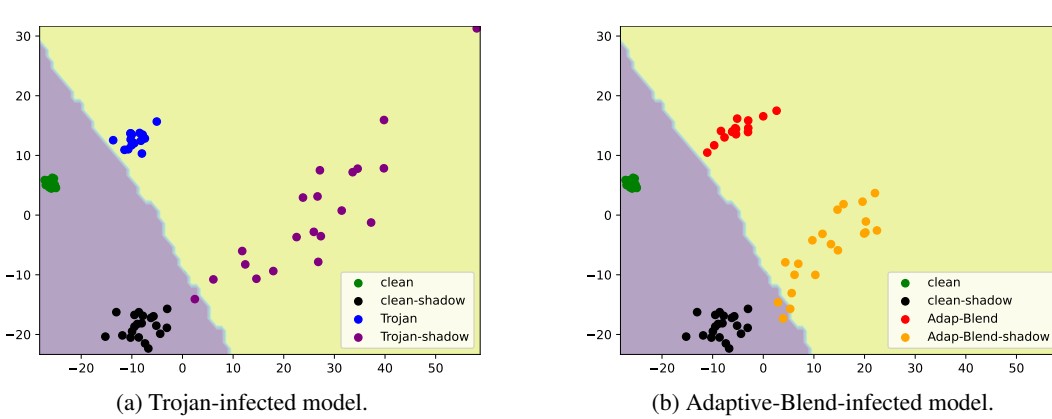

(a) Trojan-infected model.  (b) Adaptive-Blend-infected model.

Figure 5: Visualization of class subspace inconsistency through PCA.

Table 25: AUROC on Swim Transformer for CIFAR-10 and GTSRB. AVG stands for the average AUROC. Green (red) cells denote values greater (lower) than 0.8.

| | | Badnets (Gu et al., 2017) | Blend (Chen et al., 2017) | Trojan (Liu et al., 2018b) | WaNet (Nguyen & Tran, 2021) | Dynamic (Nguyen & Tran, 2020) | Adap-Blend (Qi et al., 2023b) | Adap-Patch (Qi et al., 2023b) | AVG |
|---|---|---|---|---|---|---|---|---|---|
| STRIP (Gao et al., 2019) | cifar10 | 0.998 | 0.9761 | 0.9386 | 0.483 | 0.9794 | 0.8558 | 0.8622 | 0.8704 |
| | gtsrb | 0.9851 | 0.7922 | 0.8036 | 0.5805 | 0.9999 | 0.7494 | 0.8174 | 0.8183 |
| AC (Chen et al., 2018) | cifar10 | 0.5001 | 0.5005 | 0.4999 | 0.5000 | 0.5002 | 0.4998 | 0.5001 | 0.5001 |
| | gtsrb | 0.2198 | 0.2591 | 0.5001 | 0.5012 | 0.5702 | 0.4999 | 0.5001 | 0.4358 |
| Frequency (Zeng et al., 2021) | cifar10 | 1.0000 | 0.9563 | 0.9594 | 0.5707 | 0.9999 | 0.8564 | 0.8125 | 0.8793 |
| | gtsrb | 0.9283 | 0.8835 | 0.9194 | 0.6767 | 0.8301 | 0.8531 | 0.8794 | 0.8529 |
| CT (Qi et al., 2023c) | cifar10 | 0.868 | 0.9968 | 0.9994 | 0.5142 | 0.9919 | 0.8766 | 0.9758 | 0.8890 |
| | gtsrb | 0.9125 | 0.867 | 0.9967 | 0.4279 | 0.9991 | 0.8638 | 0.7455 | 0.8304 |
| SS Tran et al. (2018) | cifar10 | 0.3877 | 0.3745 | 0.3753 | 0.2747 | 0.3749 | 0.3749 | 0.3913 | 0.3648 |
| | gtsrb | 0.4961 | 0.4925 | 0.4946 | 0.4987 | 0.4925 | 0.4925 | 0.4925 | 0.4942 |
| SCAn (Tang et al., 2021) | cifar10 | 0.9943 | 0.9943 | 0.9549 | 0.7596 | 0.7495 | 0.8451 | 0.6215 | 0.8456 |
| | gtsrb | 0.8973 | 0.6869 | 0.7661 | 0.5219 | 0.8141 | 0.6451 | 0.5577 | 0.6984 |
| SPECTRE (Hayase et al., 2021) | cifar10 | 0.5964 | 0.5981 | 0.5959 | 0.4751 | 0.5997 | 0.4751 | 0.4752 | 0.5451 |
| | gtsrb | 0.7383 | 0.4911 | 0.7383 | 0.4911 | 0.4995 | 0.4991 | 0.4911 | 0.5641 |
| BPROM (10%) | cifar10 | 1.0000 | 1.0000 | 1.0000 | 1.0000 | 0.8000 | 0.8949 | 0.8667 | 0.9374 |
| | gtsrb | 1.0000 | 1.0000 | 1.0000 | 1.0000 | 1.0000 | 0.8000 | 0.9334 | 0.9619 |

# D EXPERIMENTAL RESULTS ON IMAGENET

This section includes the experimental results on ImageNet. In particular, BPROM achieves an average AUROC of 0.9996 for ResNet18, significantly outperforming other defenses.

Table 26: AUROC of BPROM and other defense methods against various backdoor attacks on ImageNet. AVG stands for the average AUROC.

| | Badnets | Trojan | Adap-Blend | Adap-Patch | AVG |
|---|---|---|---|---|---|
| CD (Huang et al., 2023) | 0.7954 | 0.9424 | 0.6648 | 0.5842 | 0.7467 |
| SCALE-UP (Guo et al., 2023) | 0.9912 | 0.6556 | 0.3971 | 0.3339 | 0.5944 |
| STRIP (Gao et al., 2019) | 0.0500 | 0.0500 | 0.5244 | 0.5500 | 0.2936 |
| BPROM (10%) | 1.0000 | 1.0000 | 0.9986 | 0.8296 | 0.9570 |

# E NOTATION AND DEFINITIONS

For clarity and reproducibility, Table 27 summarizes the notation and definitions used throughout the paper.

Table 27: Notation and Definitions

| Symbol | Description |
|---|---|
| $D_S$ | Reserved clean dataset (1-10% of test set) |
| $D_E$ | Extracted samples from $D_S$ for poisoning |
| $D_P$ | Poisoned dataset created from $D_S$ |
| $D_T$ | External clean dataset for visual prompting |
| $D_Q$ | Random samples from $D_T$'s test set |
| $D_{\text{meta}}$ | Samples for training meta-classifier |

