# OpenReview forum: "Prompting the Unseen: Detecting Hidden Backdoors in Black-Box Models"
_ICLR.cc/2025/Conference — ICLR 2025 Conference Withdrawn Submission_

### Official Review · Reviewer_wUcE · 2024-10-19

**Soundness:** 3
**Presentation:** 2
**Contribution:** 2
**Rating:** 5
**Confidence:** 4

**Summary:**

The paper presents a new approach for detecting backdoor attacks. Specifically, the authors leverage a technique called Visual Prompting (VP) and notice that its performance degrades greatly when a model is backdoored. The authors make use of this phenomenon to detect backdoor attacks by simply applying VP to *clean* and *poisoned* models (trained by the authors) and then learning a classifier that predicts whether a model has been backdoored based on the representation it extracts from data points. The authors test their approach in a range of settings and show its efficacy.

**Strengths:**

- The paper presents an interesting approach for detecting backdoor attacks.
- The proposed approach seems to work in a range of settings.
- The authors perform a good ablation to study the effect of different hyperparameters on the defense's efficacy.

**Weaknesses:**

- Some of the examples are confusing. Specifically, Figure 1 presents a very confusing example of VP since the digit 3 is not expected to map an actual class of ImageNet. I think a better example is to choose some CIFAR-10 image that has a label in ImageNet and update the figure accordingly. (Figure 2 is also confusing.)
- The authors do not justify the choice of VP as the space where the algorithm is applied. What if the detection algorithm is applied to other spaces, e.g., the representation space of the models, etc.?
- The triggers considered are a bit large: 4x4 is a relatively big patch (for CIFAR images). It would be nice to consider smaller triggers.
- The poisoning fraction considered are also on the larger end (5%, 10%, 20%). What would happen if the poisoning ratios are smaller, e.g., 1%?
- The paper doesn't compare against MNTD, although the authors say it's the closest algorithm in its mechanics.
- The other baselines in the work detect poisoned samples, while this method only detects if a model is backdoored. This doesn't seem like a 100% fair comparison as the methods have different objectives in mind.
- The numbers do not match across tables in the paper. For example, the BPROM row from Table 5 does not match Table 23. Similarly Table 6 does not match Table 26. Can you please provide more clarity?
- The paper does not contain a *no-defense baseline*. Can you please include?

**Questions:**

- Have you considered evaluating against triggers that are very different from the ones used to learn BRPOM? For example invisible triggers?
- Have you considered evaluating against attacks that are designed to avoid class subspace inconsistency *in representation space*?

---

### Official Review · Reviewer_jCx6 · 2024-10-29

**Soundness:** 2
**Presentation:** 3
**Contribution:** 3
**Rating:** 5
**Confidence:** 3

**Summary:**

The paper presents BPROM, a novel black-box detection method for backdoor attacks in machine learning models, utilizing visual prompting (VP). The core concept of BPROM is based on class subspace inconsistency, which indicates that the accuracy of a prompted model will decrease if the source model has been compromised. This inconsistency arises from feature space distortion caused by poisoned datasets, a common characteristic of various backdoor attacks. The experimental results indicate that BPROM is effective in detecting all-to-one backdoors, where a single trigger can cause misclassification across multiple inputs.
However, it is not clear about if the experimental are done carefully, and if they are correct.
Unfortunately, there is no code added to better validate the experimental setup. However, the method faces challenges with all-to-all backdoors, where attackers can manipulate feature space distortion more effectively, making detection more complex. The authors note this limitation and propose it as an area for future research.

**Strengths:**

The paper is rich in experiments and based on this the authors could gain novel insights. On top of it, their proposed defense shows promising results even on adaptive attacks.

1. Novel Approach: BPROM introduces an innovative methodology for backdoor detection that leverages class subspace inconsistency, which is a relatively unexplored area in the context of black-box models.
2. Effective Detection: The experimental results demonstrate strong performance in identifying all-to-one backdoors, showcasing the potential of BPROM as a practical tool for enhancing model security.
3. Clear Framework: The paper provides a well-structured framework for understanding how feature space distortion affects model accuracy, contributing to the theoretical foundation of backdoor detection methods.

**Weaknesses:**

1. Limited Scope: While BPROM performs well against all-to-one backdoors, its effectiveness diminishes with all-to-all backdoors, highlighting a significant limitation in its applicability.
The reviewer thinks that based on the underlying structure of prompts, because pompts do not have so many parameters to learn, since they are just a frame around the image.

2. Future Work Needed: The authors acknowledge the need for further research to address the challenges posed by all-to-all backdoors, which may leave readers wanting more immediate solutions or insights into potential strategies.

3. Experimental Validation: The paper could benefit from a broader range of experiments to validate BPROM's effectiveness across different types of models and datasets, providing a more comprehensive evaluation of its capabilities. The current experiments show very good results and it is not clear if it is based on the authors chosen experimental setup. Unfortunately, there is no code added to this.

Writing:
For Figure 3: It does not describe how these plots are created.  Even though the message of this plot is clear to me.

**Questions:**

Experimental questions:
How did you do the plots for Figure 3? I mean, is it somewhere described? It is clear what you want to say, but as a reviewer it is hard to follow how you have achieved this plot?

Table 9: Why is there a difference in AUROC between CIFAR-10 and GTSRB dataset? For the GTSRB You have perfect AUROC values.
Table 10: also shows very perfect rates. I wonder why only F1 score and AUROC?
In overall, as reviewer, I am not sure, if there is a bug in the code. If you could add some code, it would be easier for me to understand your experiments.

More clarification would help as reviewer to change my opinion to acceptance.


Future work questions:
CIFAR-10 is a very small dataset. How do you think that it would work for more realistic datasets in the future?

Do you think that the limitation all-to-all backdoors is based on the low amount of parameters of the prompt?

---

### Official Review · Reviewer_QNS3 · 2024-11-03

**Soundness:** 1
**Presentation:** 2
**Contribution:** 2
**Rating:** 3
**Confidence:** 4

**Summary:**

The submission presents a black-box backdoor model detection method which leverages differences in the way clean and backdoored models represent images. The method exposes those differences through visual prompting - where a pretrained model is adapted to a new task by learning noise around the new dataset - because visual prompting tends to perform worse for backdoored models. The method trains and visually prompts a number of shadow clean and backdoored models on a small dataset, then trains a meta classifier on the shadow models to predict future models.

**Strengths:**

1.	Proposes using Visual Prompting as a backdoor detection technique, which appears to be a novel application of Visual Prompting
2.	Demonstrates high AUROC values across a variety of tasks.

**Weaknesses:**

1.	This submission could benefit from a clearer explanation for why visual prompting is the right tool for backdoor model detection. Improvements to Figures 1 and 2 may aid in clarifying this.
2.	This submission is lacking in detailed comparisons of other model-level detection techniques. Other methods are cited in Section 2 but not described in sufficient detail to understand how this work compares to those existing methods. While BProm was compared qualitatively to MNTD in section 5.3, there was no quantitative comparison to this similar method.
3.	This method is presented as a black-box backdoor detection method yet for almost all evaluations assumes that the shadow models are trained with the same model architecture and training data distribution as the target model. This paper should more concretely describe the role that these similar model architectures and training data distributions play in the success of this method and evaluate the performance without these knowledge assumptions.
4.	This submission leaves out several important evaluation details such as:
a) Hyper-parameter q (the size of D_Q) is introduced but never specified (line 282)
b) The gradient-free optimization method used is never specified (line 274). An example algorithm is given, but there is no indication that is the one used.
c) Default backdoor parameters are never stated. As one example, what is the poison rate in Table 3, and what is the trigger size in Table 4?
d) It is not stated what the backdoor technique for poisoning the shadow models is. Importantly, it is unclear whether the method assumes knowledge of the target model's poisoning method.
5.	When comparing against MNTD, the authors state that a benefit of BProm over MNTD is that BProm uses a single backdoor attack instead of multiple. They state that using multiple methods only marginally improves detection accuracy. They should quantitatively show the impact of using single backdoor attacks versus multiple and clearly state the assumptions about the diversity of types of backdoor attacks used for evaluation of both BProm and MNTD.
6.	There are some inconsistencies in results reported in the text versus in the tables. If the text and tables are referring to different quantities, please clarify in the text. Examples of these inconsistences are:
a) Line 321 claims that BPROM achieves "0.8137 F1-score on CIFAR-10 with BadNets and STL-10, and 0.7499 with GTSRB and STL-10", however Table 16 indicates that BPROM achieves an F1-score of 1.0 on both datasets.
b) Line 424 claims Table 6 shows that "BPROM achieves an average AUROC of 0.899 for ResNet18 and 0.912 for MobileNet", however, the table shows 0.979 for ResNet18 and 0.992 for MobileNet.
c) Section D claims that BPROM achieves an average AUROC of 0.9996 for ResNet18 on ImageNet, however Table 26 shows an average of 0.9570.
7.	There is inconsistency in when attacks and defenses are evaluated and rationale is not given for the omissions. For instance:
a) The experiments comparing trigger size and poisoned rates (Tables 3, 4, 8,9) are only performed on Blend and Adap-Blend attacks but results in Tables 5 and 6 are reported across a greater variety of attacks.
b) Table 5 does not evaluate against CD or SCALE-UP, while Table 6 does not evaluate against Frequency, SentiNet, SPECTRE, or TED.
8.	Attacks that are described as adaptive in section 6.4 represent different settings of attacks, not specifically attacks adapted against the described defense. For instance, the low poison rates could’ve been explored in Table 9 and  clean label attacks could’ve been included amongst the other attacks in Tables 5 and 6.
9.	Submission does not discuss limitations to the model in detail. To name some:
a) Section 6.3 says that "our detection method's AUROC remains stable, with minor fluctuations", despite having very poor AUROC for CIFAR-10 and poison rate of 5%.
b) Conclusion notes that the method struggles with all-to-all backdoors because the "feature space distortion is more controllable by the attacker". No elaboration is given, especially on why all-to-one backdoors would not be just as controllable.

**Questions:**

Please address the concerns detailed above.

---

### Note · Authors · 2024-11-13

I have read and agree with the venue's withdrawal policy on behalf of myself and my co-authors.